# DERD: Data-free Adversarial Robustness Distillation through Self-adversarial Teacher Group

## ABSTRACT

Computer vision models based on deep neural networks are proven to be vulnerable to adversarial attacks. Robustness distillation, as a countermeasure, takes both robustness challenges and efficiency challenges of edge models into consideration. However, most existing robustness distillations are data-driven, which can hardly be deployed in data-privacy scenarios. Also, the trade-off between robustness and accuracy tends to transfer from the teacher to the student, and there has been no discussion on mitigating this trade-off in the data-free scenario yet. In this paper, we propose a Data-free Experts-guided Robustness Distillation (DERD) to extend robustness distillation to the data-free paradigm, which offers three advantages: (1) Dual-level adversarial learning strategy achieves robustness distillation without real data. (2) Expert-guided distillation strategy brings a better trade-off to the student model. (3) A novel stochastic gradient aggregation module reconciles the task conflicts of the multi-teacher from a consistency perspective. Extensive experiments demonstrate that the proposed DERD can even achieve comparable results to data-driven methods.

## CCS CONCEPTS

• **Computing methodologies** → *Computer vision representations.*

## KEYWORDS

Data-free, Adversarial Robustness, Knowledge Distillation

## 1 INTRODUCTION

Computer vision (CV) systems relying on deep neural networks (DNNs) demonstrate outstanding performance across various tasks, including image classification [22], object detection [33], and person ReID [41, 46, 51]. However, recent studies indicate that DNNs are vulnerable to adversarial attacks, which involve the addition of carefully hand-crafted perturbations to the input. These perturbations lead to the complete deception of DNNs, resulting in incorrect decisions [16, 27, 35]. This vulnerability presents challenges to the reliable deployment of DNN-based systems.

As countermeasures, various adversarial defense mechanisms have been proposed [23, 32, 39, 50]. Adversarial Training (AT) [16, 20, 27] is one of the most effective defense strategies. By integrating adversarial examples into the training process dynamically, the model learns robust representations through a min-max game

**Unpublished working draft. Not for distribution.**

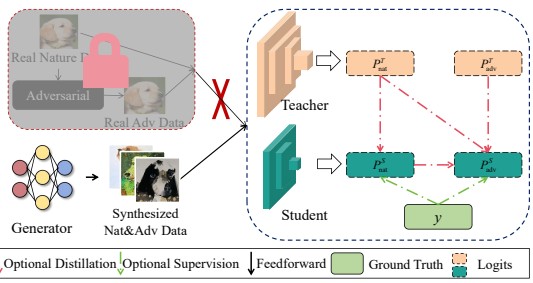

**Figure 1: Illustration of data-free robustness distillation. In data-driven scenarios, the student model can align with the teacher's mapping preference by utilizing real adversarial examples. However, in data-privacy scenarios, accessing real natural samples is often impractical and thus it is impossible to obtain real adversarial examples crafted from benign data. Thus, the pseudo data is required.**

to counter potential attacks. However, the effectiveness of AT for lightweight models is hindered due to their limited capacity and representation ability. Moreover, the deployment of large-scale adversarially trained models in practical applications is often challenging due to requirements for timeliness and memory. To address these challenges, robustness distillation [14, 52, 53] has been introduced. This approach tackles both robustness and efficiency concerns by employing a larger robust model (teacher model) to guide the robustness training of a smaller model (student model).

However, most existing robustness distillation techniques are data-driven [12] methods, which may pose challenges in practical deployment scenarios where data privacy protection and transmission efficiency are crucial. Firstly, these methods often assume continuous access to the real training data throughout adversarial training and robust distillation processes. However, some sensitive and private data (e.g., facial data, pedestrian data, and patients' medical records) may become inaccessible once the model is published to the public. Secondly, some approaches rely on access to proxy or auxiliary data beyond the real data [3, 11]. Unfortunately, these methods not only require additional training data, but also suffer from performance degradation when the distribution of the proxy data differs from that of the original data. In summary, existing robustness distillation methods still face challenges for data privacy.

The analyses above emphasize the crucial need to extend robustness distillation to scenarios where real data is unavailable. Data-free Knowledge Distillation [5, 12, 13, 25, 28, 42] offers valuable insights that, the inherent knowledge of a teacher model can be effectively transferred to a student model by leveraging artificially constructed pseudo data. In this approach, the teacher model guides the generation of pseudo data (no matter explicit or implicit), eliminating the necessity for real data. Similarly, the robust knowledge of

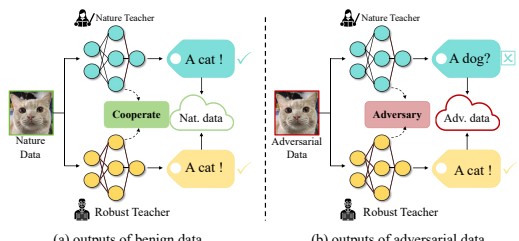

(a) outputs of benign data    (b) outputs of adversarial data

**Figure 2: An intuitive description of the preference (bias) of natural and adversarial samples. Ideally, both the natural and robust teachers can correctly classify natural data. However, adversarial data, crafted to deceive models, is typically only correctly classified by the robust teacher. This contrast inspires the creation of pseudo adversarial data, which should only be correctly classified by the robust teacher while leading to misclassification of the natural teacher, effectively capturing the nuanced preference (bias) of the two models.**

a robust teacher model can be transferred to a student model using the 'generation-distillation' paradigm, as illustrated in Figure 1.

To our best knowledge, DFARD [37] is the pioneering and only work addressing the data-free robustness distillation task. However, it proposed only several trivial tricks without fully considering the unique challenges inherent in this task. Firstly, existing data-free distillation methods, including DFARD, primarily focus on mining trivial discriminative knowledge (i.e., nature knowledge) from the teacher, rather than extracting the robustness knowledge in an adversarial min-max game. Generating adversarial samples is rather difficult in the absence of authentic benign samples. Secondly, the common trade-off between robustness and accuracy [45] may be inadvertently transferred from the teacher to the student during robustness distillation. While some methods attempt to mitigate this trade-off [45, 49], none of them addresses it in data-free scenario. Optimizing the trade-off in data-privacy scenario demands that defenders to run out of the dependency on real data and find another way out move beyond reliance on real data and explore alternative solutions, such as leveraging gradients. DFARD lacks a min-max process similar to data-driven robustness distillation, and also fails to address the sub-optimal solution resulting from the trade-off.

In this paper, we propose data-free experts-guided robustness distillation (DERD) to address the aforementioned challenges. Firstly, we propose a 2-stage distillation framework. In stage-I, the student model learns natural knowledge from a natural teacher, serving as a warm-up phase, wihch provides a solid pre-training for subsequent robust learning processes [34]. In stage-II, we craft pseudo adversarial examples by maximizing the output discrepancy between the natural teacher and the adversarial-defended robust teacher. The fundamental concept of stage-II is illustrated in Fig. 2, where the pseudo data is engineered to deceive the nature teacher while simultaneously maximizing the output descrepancy between the nature and robust teachers. Essentially, this stage forms a dual-level adversarial learning mechanism, involving (1) the adversarial mechanism between teachers and generators, and (2) the self-adversarial mechanism between the natural and robust teachers. Secondly, to mitigate the trade-off between accuracy and robustness, we employ a homogenized expert-guiding strategy, where both natural

knowledge and robust knowledge are distilled from the natural teacher and robust teacher respectively, using the same surrogate data. Lastly, we introduce a stochastic gradient aggregation (SGA) module to harmonize the gradient of both natural and robust distillation tasks. This module optimizes the 'robustness-accuracy' trade-off by ensuring consistency from a gradient perspective.

Our main contributions can be summarized as follows:

(1) We propose a novel data-free robustness distillation method. Comparing to the only existing solution, we design a tailored 2-stage framework aimed at extracting robust knowledge through a min-max game, similar to data-driven defense strategies.

(2) To balance the inherent trade-off between robustness and accuracy in data-free manner, we introduce an expert-guiding strategy and employ the SGA regularizer to reconcile this optimization conflict from both sample level and gradient level.

(3) Our approach demonstrates better performance on mainstream evaluation datasets compared to the only one previous data-free robustness distillation method. Furthermore, it is also comparable to the data-driven distillation methods.

## 2 RELATED WORKS

### 2.1 Knowledge Distillation

Knowledge distillation is a technique aimed to transfer knowledge from a large model to a more efficient and smaller model. It can be traced back to decision trees, where a decision tree is trained to mimic the output of multiple decision trees [1]. Hinton *et al.* extended this idea to neural networks and termed it as 'knowledge distillation' [18]. In this approach, a compact student model learns the mapping relationship from a large, high-performance teacher model. Over time, the introduction of various variants and training techniques [30] has enabled model compression through knowledge distillation in many fields [6, 47].

The assumption of 'data availability' in vanilla knowledge distillation overlooks the more practical scenario of 'data unavailability.' Recent research has begun to address this gap by focusing on data-free knowledge distillation methods, which is promising and draws lots of attention. For instance, Lopes *et al.* [26] synthesize inputs based on pre-stored auxiliary layer-wise statistics (meta-data) of the teacher model. Chen *et al.* [26] train a generator for image generation while treating the teacher model as a fixed discriminator. ADI [42] utilizes batch normalization statistics (BNS) from a pre-trained teacher to optimize input noise for generating high-quality images. CMI [13] leverages local and global contrast of samples to optimize the diversity of the generator. ZSKT [28] employs adversarial distillation, transferring knowledge from teacher to student using KL divergence and spatial attention, while DFAD [12] solely utilizes MAE loss to perform the min-max process between teacher and student models for better alignment. However, these methods aim to extract benign discriminative knowledge, and directly incorporating it into robustness distillation won't be the most effective approach.

### 2.2 Adversarial Attack & Defense

Adversarial attacks aim to deceive the target model by introducing minor perturbations to the inputs. These attacks are categorized based on the level of access the attacker has to the target

model,resulting in two main categories: white-box and black-box attacks. In white-box attacks, the attacker has complete access to the target model, including gradients and parameters. Mainstream white-box attack methods include gradient-based approaches [8, 16, 21, 27], classifier-based methods [29], and optimization-based techniques [4]. On the other hand, black-box attacks assume limited prior information about the target model. These attacks are further classified into score-based attacks, decision-based attacks, and transfer-based attacks. Decision-based attacks operate under the constraint that the attacker can only access the one-hot hard labels from the target model. For instance, the boundary attack [2]. Score-based attacks, such as ZOO [9], enable the attacker to obtain probability scores of the input queries, offering more detailed information beyond the final decision. Transfer-based attacks [9, 24, 40, 48, 54] involve building a proxy model of the target model, commonly used to evaluate the black-box adversarial robustness of DNNs.

Adversarial defenses aim to maintain the robustness of DNNs against adversarial attack. Early heuristic defense methods, while reporting promising results, have been found to rely on 'obfuscated gradients', rendering their unreliable robustness. Adversarial training (AT) [16, 20, 27] is considered one of the most effective defenses. However, the effectiveness of AT for small models is constrained by their limited capacity. To address this, robustness distillation [14, 52, 53] is proposed, aiming to transfer robustness from a large, robust model to a more efficient, smaller model. ARD [14] and IAD [52] have demonstrated that robustness distillation can yield a student network with greater robustness than training from scratch. RSLAD [53] introduces the concept of robust soft labels (RSL) provided by the robust teacher, which can offer an effective robust representation for the student. Additionally, MTARD [49] proposes a dual-teacher structure to optimize the trade-off between accuracy and robustness in robustness distillation. Furthermore, Trades [45] and MART [38] can also be considered examples of robust self-distillation, as they leverage the model's outputs on natural samples to guide its outputs on adversarial samples.

However, there have been few works to address robustness distillation in scenarios where real data is unavailable. To our best knowledge, only two works have explored data-free robustness distillation: DFHL-RS [43] and DFARD [37]. However, DFHL-RS primarily focuses on model stealing attacks rather than robust distillation and operates in a completely black-box setting. On the other hand, DFARD only introduces some basic trivial training techniques for all distillation tasks, without considering how to efficiently utilize the robust knowledge inherent in robust teachers. In contrast, our approach achieves superior data-free adversarial robustness through a tailored framework.

## 3 METHODOLOGY

### 3.1 Preliminaries

**Adversarial attack (untargeted).** Given a target model $f_w$ parameterized by $w$, the objective of the attacker can be formulated as a conditional optimization problem:

$$\arg\max_{x'} \mathcal{L}(f_w(x', w), y), \text{ s.t. } \left\| x' - x \right\|_p \leq \varepsilon, \quad (1)$$

where $x'$ denotes the adversarial examples, $\mathcal{L}(\cdot)$ represents the classification loss (e.g., cross entropy loss), and $\varepsilon$ is the upper bound of the perturbation under the $l_p$-norm. The goal of the attacker is to deceive the target model with visually imperceptible perturbations.

**Adversarial defense** aims to preserve the discriminability capability of $f_w$ under adversarial attacks, i.e., achieving adversarial robustness. The objective of defense can be formalized as:

$$\arg\min_{w} \mathcal{L}(f_w(x', w), y). \quad (2)$$

**Robustness distillation** addresses both robustness and efficiency challenges. Given a student model $S_{\theta_S}(\cdot)$ (abbreviated as $S(\cdot)$) and a teacher model $T_{\theta_T}(\cdot)$ (abbreviated as $T(\cdot)$), the main goal of robustness distillation is to transfer the robustness of the teacher model against adversarial examples to a smaller student model:

$$\arg\min_{\theta_S} \mathcal{L}(S(x, \theta_S), y) + \mathcal{D}(S(x'), T(x')), \quad (3)$$

where $\mathcal{D}(\cdot)$ denotes the discrepancy in output distribution between the teacher and student models, typically measured using metrics like KL divergence.

**Data-free robustness distillation** refers to scenarios that the real data $x$ is inaccessible, making the real adversarial examples $x'$ based on $x$ also unavailable. Consequently, the defender must generate substitute data and align the mapping relationship between the teacher and student based on these pseudo data, presenting additional challenges. This process can be formalized as:

$$\arg\min_{\theta_S} \mathcal{D}(S(G(z)), T(G(z))), \quad (4)$$

where $G(\cdot)$ represents the generator, which can be either explicit forms (such as generative networks [15]) or implicit forms (such as model inversion [42]). Here, $z$ denotes the random input to the generator, typically sampled from a Gaussian distribution.

Our proposed *Data-free Expert-guided Robustness Distillation* (DERD) is a dual-stage model based on an explicit generator which also incorporates a regularizer, SGA, to reconcile the trade off between accuracy (Acc.) and robustness (Rob.), shown in Fig. 3. The in stage-I and stage-II (including the regularizer) will be introduced in Sec 3.2 and Sec 3.3 respectively.

### 3.2 Stage-I: Warm-up

In stage-I, we initialize the student model $S(\cdot)$ and generator $G(\cdot)$ using a natural teacher $T_{nat}(\cdot)$. To clarity, the objective function in the stage-I primarily consists of the loss function of the generator $\mathcal{L}^G_{stage-I}$ and the loss function of the student model $\mathcal{L}^S_{stage-I}$:

$$\mathcal{L}_{stage-I} = \mathcal{L}^G_{stage-I} + \mathcal{L}^S_{stage-I}. \quad (5)$$

The optimization objective of $G(\cdot)$ is to generate substitute data for real data, facilitating the distillation of natural knowledge to the student. By pairing a pre-trained teacher with a generative network $G$, a GAN-like framework for adversarial learning [5] is formed, enabling the generator to produce pseudo-natural data, where the teacher acts as the discriminator $D$. However, there are key differences from a vanilla GAN: (1) The teacher model is frozen as a supervisor $D$. (2) The role of the teacher model is no longer to determine the authenticity of images but rather to classify the data generated by $G$ into different conceptual sets, transitioning from a binary classification task to a multi-class classification task.

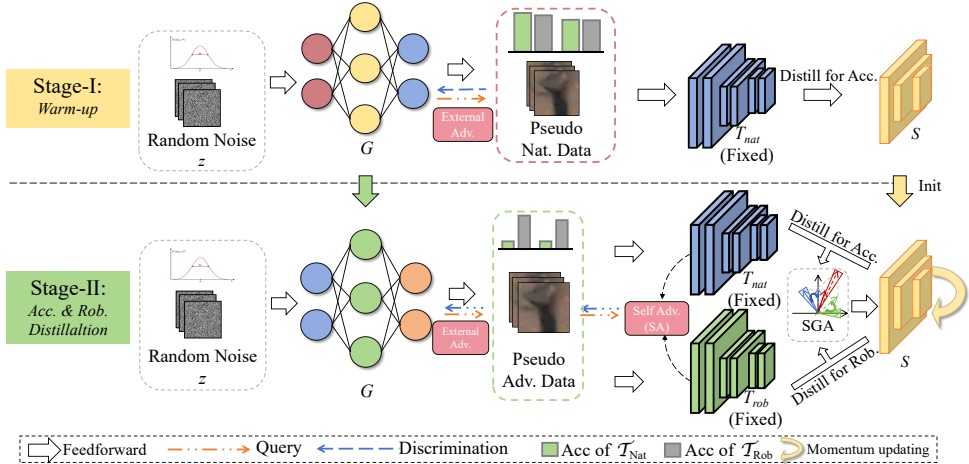

**Figure 3: Model details of our DERD. In stage-I, only $T_{nat}$ is used to supervise the warm-up process of both $G$ and $S$. In stage-II, we introduce two key components: the self-adversary (SA) module based on the experts-guiding (EG) strategy and the stochastic gradient aggregation (SGA) module. The external adversarial component consists of the generator and the teacher group ($T_{nat}$ and $T_{rob}$), while the self-adversarial strategy only involves the teachers.**

Essentially, the teacher model provides supervisory information for training the generator $G$.

Based on the above analysis, the primary loss of $G$ in a mini-batch can be formalized as:

$$\mathcal{L}_{\text{oh}} = \frac{1}{n}\sum_{i}^{n}\mathcal{L}_{\text{cross}}\left(T_{nat}(G(z^i)), \hat{y}^i\right), \quad (6)$$

where $\mathcal{L}_{oh}$ represents the one-hot classification loss, $z^i$ denotes a stochastic input, $\hat{y}^i$ represents a pseudo target label, and $n$ is the number of pseudo samples in a minibatch. This loss function aims to train $G(\cdot)$ to generate pseudo data from random inputs, ensuring that the pseudo data $G(z^i)$ is correctly classified by $T_{nat}$ (i.e., the discriminator) with the corresponding pseudo target labels $\hat{y}^i$.

In addition to direct discriminative supervision, a common assumption is that if the generated data are discriminative, their patterns should be significantly captured by the filters in the network, leading to higher activation values in the intermediate layers. Consequently, a prior loss based on feature activation values is:

$$\mathcal{L}_a = -\frac{1}{n}\sum_{i}^{n}\left\|Feat_{nat}(G(z^i))\right\|_1, \quad (7)$$

where $Feat_{nat}(G(z^i))$ denotes the hidden layer features before the classifier of $T_{nat}$, and the objective is for $G(\cdot)$ to generate pseudo data with rich features rather than sparse trivial solutions.

Besides, we aim to fully leverage the statistical prior within the teacher model, as it is trained on real data. To achieve this, we propose minimizing the discrepancy between the feature statistics of real data (indirectly encoded in the batch normalization layers of the teacher model) and pseudo data:

$$\mathcal{L}_{\text{BN}}(G(z)) = \sum_{l}\left\|\mu_l(G(z)) - \mathbb{E}\left(\mu_l(x)\mid \mathcal{X}\right)\right\|_2 +$$
$$\sum_{l}\left\|\sigma_l^2(G(z)) - \mathbb{E}\left(\sigma_l^2(x)\mid \mathcal{X}\right)\right\|_2, \quad (8)$$

where $\mu_l(\cdot)$ and $\sigma_l(\cdot)$ represent the mean and variance of the input data in the $l^{th}$ layer, respectively, and $\mathcal{X}$ represents the distribution of real data, where $x \in \mathcal{X}$. This technique was initially employed in data-free distillation using model inversion [42], but its application in generator-based approaches has been scarcely discussed.

Moreover, to eliminate the model's bias for certain categories, it's typically desirable for the target label distribution of the conceptual set to be balanced. This implies that the sample quantity and occurrence probability for each category should be consistent. Given a set of output vectors for the pseudo data $\{T_{nat}(G(z^1)), T_{nat}(G(z^2)), \cdots, T_{nat}(G(z^n))\}$, the count for a specific class $c$ is $y_T^c = \sum_{i=1}^{n}\delta(T_{nat}(G(z^i)) = c)$. Here, $\delta(\cdot)$ is an indicator function:

$$\delta(A) = \begin{cases} 1, & \text{if } A \text{ is true} \\ 0, & \text{if } A \text{ is false.} \end{cases} \quad (9)$$

Then, the information entropy loss of $G(\cdot)$ can be expressed as:

$$\mathcal{L}_{ie} = -\mathcal{H}_{\text{info}}\left(\frac{1}{n}\sum y_T^c\right), \quad (10)$$

where $\mathcal{H}_{\text{info}}(p) = -\frac{1}{k}\sum_{i} p_i \log(p_i)$. This loss aims to maximize the entropy of the distribution of the generated classes. When the generator produces each class with equal probability, Eq. 10 is minimized.

Therefore, the loss function for the generator in the stage-I can be summarized as:

$$\mathcal{L}_{stage1}^G = \lambda_{oh}\mathcal{L}_{oh} + \lambda_a\mathcal{L}_a + \lambda_{\text{BN}}\mathcal{L}_{\text{BN}} + \lambda_{ie}\mathcal{L}_{ie}. \quad (11)$$

Based on the pseudo data $G(z)$, the optimization goal of the student is to mimic the mapping relationship of the teacher model:

$$\mathcal{L}_{stage-I}^S = \frac{1}{n}\sum_{i}^{n}\mathcal{KL}\left(S(G(z^i)), T_{nat}(G(z^i))\right), \quad (12)$$

where $\mathcal{KL}$ is the KL-divergence. After stage-I, the student $S(\cdot)$ initially obtains the natural knowledge, which can be used as the

initialization for stage-II. Additionally, $G(\cdot)$ acquires the ability to generate pseudo natural data. As will be discussed below, pseudo data for adversarial samples can be obtained based on $G(z)$.

### 3.3 Stage-II: Rob & Acc Distillation

In stage-II, we employ the nature teacher $T_{nat}$ and robust teacher $T_{rob}$ for self-adversarial learning and expert strategy to distill natural and robust knowledge to the student model.

• **Self-adversary (SA).** The primary task of stage-II is to find substitute data for adversarial samples to explore the robustness of the robust teacher $T_{rob}$. As illustrated in Fig. 2, a natural characteristic of adversarial samples is to maximize the discrepancy between the robust and natural teachers. Utilizing $G(z)$ as substitute data for natural data, we construct *discrepancy data* $x_d$ based on $G(z)$ as the pseudo adversarial data, to maximize the output discrepancy between $T_{nat}$ and $T_{rob}$ in a self-adversary manner:

$$x_d^t = G(z), \quad \text{if } t = 0, \tag{13}$$

$$x_d^{t+1} = (x_d^t + \alpha \cdot \text{sgn}\left(\nabla_{x_d^t} D(T_{nat}(x_d^t), T_{rob}(x_d^t))\right), \quad \text{if } t > 0, \tag{14}$$

where $t$ represents the iteration step. In data-free distillation, L1 loss is considered a better metric for measuring discrepancy compared to KL divergence [12, 36]. This preference arises because the gradient of KL divergence tends to be smaller than that of L1 loss, making it more susceptible to gradient vanishing. Therefore, we choose the L1 norm to quantify the output difference between $T_{nat}$ and $T_{rob}$:

$$D(T_{nat}(x_d^t), T_{rob}(x_d^t)) = \mathbb{E}_{z \sim p_z(z)} \|T_{nat}(x_d^t) - T_{rob}(x_d^t)\|_1. \tag{15}$$

Clearly, $x_d$ is obtained through an iterative process. Note that the optimization process is not bound by the norm constraint, as the pseudo data do not require visual similarity. The discrepancy metric enables the pseudo-data to search for samples in the input space that can deceive the natural teacher. However, solely adopting this objective may be suboptimal, as the pseudo data may tend to deceive the robust model rather than the natural model to maximize $D(T_{nat}, T_{rob})$. Hence, we introduce an entropy constraint in the optimization objective to ensure that the pseudo data can indeed deceive the natural teacher while maximizing the discrepancy:

$$x_d^{t+1} = x_d^t + \alpha \cdot \text{sgn}\left(\nabla_{x_d^t}(D(T_{nat}(x_d^t), T_{rob}(x_d^t)) + \lambda_d \mathcal{L}_{ce}(x_d^t, T_{nat}(x_d^t)))\right),$$
$$\text{if } t > 0. \tag{16}$$

• **Expert-guiding strategy (EG).** Using the discrepancy data $x_d$, we employ the expert-guiding strategy to simultaneously distill both natural knowledge and robust knowledge into $S$. In MTARD [49], where the data is available, natural knowledge is distilled using natural data, while robust knowledge is distilled using adversarial examples. However, we find that utilizing $x_d$ for the nature teacher can also enhance the distillation of natural knowledge. This may be because $x_d$ can serve as challenging samples for the natural teacher, further leveraging the natural knowledge. The expert distillation strategy for the student model can be formalized as:

$$\mathcal{L}_{Dis} = \mathcal{KL}(T_{nat}(x_d), S(x_d)) + \lambda_{rob} \mathcal{KL}(T_{rob}(x_d), S(x_d)). \tag{17}$$

• **Gradient aggregation.** Intuitively, the loss function of the expert strategy involves two optimization tasks: distillation of natural knowledge and robust knowledge. However, directly optimizing

these two tasks with the gradient descent algorithm might be unco-ordinated, as the gradients of the two losses may not align well. The intuitive representation, as shown in Fig. 4, is that two optimization directions forming an obtuse angle could lead to a suboptimal aggregated direction. We verify this hypothesis in Fig. 5. In CIFAR10 and CIFAR100, the gradients of the robustness and natural distillations consistently form an obtuse angle, resulting in a sub-optimal joint optimization of the two tasks.

A similar issue has also been raised in unsupervised domain adaptation (UDA), where the domain adaptation loss and the classi-fication loss are often have uncoordinated aggregation directions. To address this, a gradient aggregation (GA) strategy [19] has been proposed to harmonize the two tasks. GA can be formalized as:

$$g = \left(1 - \delta\left(g_1^T g_2 < 0\right) \frac{g_2^T g_1}{\|g_1\|^2}\right)g_1 + \left(1 - \delta\left(g_1^T g_2 < 0\right) \frac{g_1^T g_2}{\|g_2\|^2}\right)g_2, \tag{18}$$

where $g_1$ and $g_2$ represent the gradient of the two losses, and $\delta(\cdot)$ is the indicator as defined in Eq. (9). For convenience, we have:

$$\tau_1 = 1 - \delta\left(g_1^T g_2 < 0\right) \frac{g_2^T g_1}{\|g_1\|^2},$$
$$\tau_2 = 1 - \delta\left(g_1^T g_2 < 0\right) \frac{g_1^T g_2}{\|g_2\|^2}. \tag{19}$$

Hence the aggregated gradient can be simplified as:

$$g = \tau_1 g_1 + \tau_2 g_2, \tag{20}$$

and the GA loss can be simplified as:

$$\tilde{L} = \int (\tau_1 g_1 + \tau_2 g_2)\, d\theta = \tau_1 L_1 + \tau_2 L_2. \tag{21}$$

The gradient harmonization process described above can be in-tuitively represented by Figs. 4 (a) and 4 (d). Gradient aggregation (GA) does not intervene when the gradients of the two loss func-tions form an acute angle. However, for two loss functions whose gradients form an obtuse angle, GA calculates the orthogonal basis for the two gradient directions respectively, and then employs this orthogonal basis to achieve a more efficient gradient aggregation.

• **Stochastic gradient aggregation (SGA).** Based on the GA module, we further propose a SGA strategy. Our objective is to in-troduce minor perturbations to the two original gradients, thereby exploring richer and more efficient aggregation directions and en-hancing the model's robustness to gradient augmentation. This approach is based on a simple intuition: by augmenting at the gradient level, we can implicitly achieve data-level augmentation, thereby improving the richness and noise resistance of gradient aggregation [51]. Adding subtle gradient perturbations to both can be formalized as:

$$g = \tau_1(g_1 + r_1) + \tau_2(g_2 + r_2), \tag{22}$$

where $r_1$ and $r_2$ are two minor stochastic gradient perturbations. Integrating Eq. 22 with respect to $\theta_S$ results in a loss for SGA:

$$\mathcal{L}_{SGA} = \int (\tau_1(g_1 + r_1) + \tau_2(g_2 + r_2))\, d\theta_S$$
$$= \tau_1 L_1 + \tau_2 L_2 + (\tau_1 r_1 + \tau_2 r_2)\theta_s \tag{23}$$
$$= \tau_1 L_1 + \tau_2 L_2 + (\tau r)\theta_s,$$

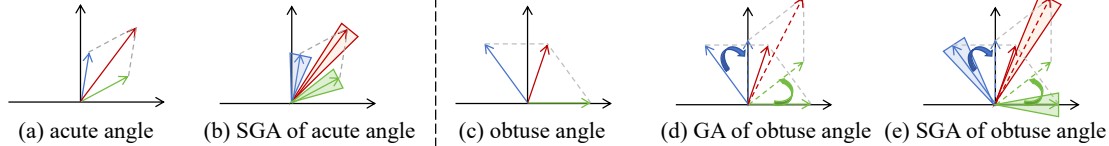

(a) acute angle  (b) SGA of acute angle | (c) obtuse angle  (d) GA of obtuse angle (e) SGA of obtuse angle

**Figure 4: An explanation of gradient aggregation. When the gradients of two tasks form an acute angle (as shown in (a)), SGA can be directly applied without the need for harmonization (as shown in (b)). However, when the gradients of the two tasks form an obtuse angle (as shown in (c)), it become necessary to perform gradient harmonization through GA (as illustrated in (d)), before aggregating the stochastic augmented gradients by SGA (as depicted in (e)).**

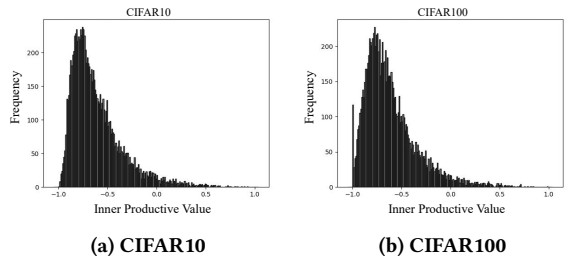

(a) CIFAR10     (b) CIFAR100

**Figure 5: The conflict between the gradients of natural knowledge distillation and robust knowledge distillation. Specifically, the gradients of natural knowledge and robust knowledge consistently form obtuse angles, as evidenced by cosine values less than 0. This conflict leads to suboptimal robustness and accuracy concurrently.**

where $\tau$ is a hyper-parameter controlling the gradient perturbation, and $r$ is the stochastic perturbation composed of $r1$ and $r2$. Therefore, the stochastic gradient aggregation can be realized through such a simple regularizer. To maintain training stability, similar to SGP [51], we introduce an L2 norm constraint to the regularizer:

$$\mathcal{L}_{SGA} = \tau_1 L_1 + \tau_2 L_2 + \tau||r\theta_s||_2. \qquad (24)$$

where $L_1$ and $L_2$ represents the original task, and the regularizer can be simplified as:

$$\mathcal{L}_{SGA} = \tau||r\theta_s||_2. \qquad (25)$$

The regularizer can be considered as a stochastic extension form of L2 regularization. On one hand, it allows the student model to explore richer gradient aggregation directions. On the other hand, it imposes the norm constraint of the student's parameters, providing regularization from the perspective of smoothness and sparsity.

• **Overall loss stage-II**. For $G(\cdot)$, the loss functions in stage-I and stage-II remain consistent (Eq. (10)), primarily generating natural alternative data $G(z)$ that is more distinguishable by $T_{nat}$.

$S(\cdot)$ learns both natural and robust knowledge simultaneously from the teacher group through the $x_d$ based on $G(z)$, and the total loss in stage-II can be formalized as:

$$\mathcal{L}^S_{stage-II} = \mathcal{L}_{Dis} + \lambda_{SGA}\mathcal{L}_{SGA}. \qquad (26)$$

**Table 1: Evaluation on the teacher models.**

| Datasets | Teachers | Backbone | Clean | FGSM | PGD$_S$ | PGD$_T$ | CW | AA | Ave |
|----------|----------|----------|-------|------|---------|---------|-----|-----|-----|
| CIFAR10 | $T_{nat}$ | RN34 | 92.77 | 12.46 | 6.71 | 5.74 | 6.27 | 0.44 | 20.73 |
| | $T_{rob}$ | RN34 | 70.89 | 69.34 | 56.78 | 56.80 | 58.42 | 42.14 | 59.06 |
| CIFAR100 | $T_{nat}$ | RN50 | 70.63 | 6.49 | 4.75 | 5.12 | 5.92 | 3.69 | 16.09 |
| | $T_{rob}$ | WRN3420 | 54.17 | 43.05 | 31.94 | 32.10 | 30.94 | 26.74 | 36.49 |
| ImageNet100 | $T_{nat}$ | ViT$_s$ | 89.49 | 52.10 | 6.15 | 7.32 | 2.54 | 0.00 | 26.26 |
| | $T_{rob}$ | ViT$_s$ | 78.44 | 66.79 | 50.01 | 53.68 | 66.17 | 55.62 | 61.78 |

## 4 EXPERIMENTS

### 4.1 Experiments Setup

**Datasets and backbones.** We evaluate our DERD on two CIFAR datasets commonly used for adversarial attack and knowledge distillation, and also discuss DERD's performance on a relatively larger dataset ImageNet100.

• CIFAR10. We employ ResNet34 [17] as the backbone for both the natural and robust teacher models, and ResNet18 & MobileNet2 [31] for the student model.

• CIFAR100. ResNet50, Wide-ResNet-3420 [44], and ResNet18 & MobileNet2 are selected as the backbones for the natural teacher, robust teacher, and student, respectively.

• ImageNet100. ViT-small [10] is chosen for the natural and robust teachers, and ViT-tiny for the student.

Our backbone selection is based on previous works in robust distillation [45, 53] and data-free distillation [5, 12, 42]. Additionally, we consider specific factors for each dataset. Specifically, we verify our model's feasibility within a homogenous teacher group (where the teachers share the same backbone) on CIFAR10, ascertain its performance with a heterogeneous teacher group (where the teachers own different backbones) on CIFAR100, and validate its applicability on ViT-based models using ImageNet100.

**Attacks.** We assess the student' performance against five commonly used the white-box attacks: FGSM [16], PGD$_S$ [27], PGD$_T$ [45], CW [4], and AutoAttack [7]. For CIFAR10 and CIFAR100, we set the $L_\infty$ norm attack budget $\epsilon = 8/255$, perturbation step size $\eta_1 = 2/255$, number of iterations $K = 10$, and batch size $m = 512$. For ImageNet100, We set $\epsilon = 0.03$, $\eta_1 = 2/255$, $K = 10$, and $m = 128$. Additionally, besides white-box attacks, we also briefly evaluate the student's robustness against black-box attacks on CIFAR10, including transfer-based and query-based black-box attacks.

**Details.** We employ the SGD optimizer with a momentum of 0.9 and weight decay of 5e-4 to train both the student and teacher models. The natural teacher and the robust teacher are trained over 2000

 

**Table 2: White-box robustness evaluation on ResNet18 for CIFAR10 and CIFAR100. The best achieved through data-free methods are highlighted in bold. MSA denotes the Model-stealing-Attack, which has similar settings to DFARD. '−' means the $PGD_T$ is not evaluated by DFHL_RS.**

| | Models | CIFAR10 | | | | | | | CIFAR100 | | | | | | |
|---|---|---|---|---|---|---|---|---|---|---|---|---|---|---|---|
| | | Clean | FGSM | $PGD_S$ | $PGD_T$ | CW | AA | Ave | Clean | FGSM | $PGD_S$ | $PGD_T$ | CW | AA | Ave |
| | Nature | 94.65 | 19.26 | 0.0 | 0.0 | 0.0 | 0.0 | 18.98 | 75.55 | 9.48 | 0.0 | 0.0 | 0.0 | 0.0 | 14.17 |
| Data Driven | SAT [27] | 83.38 | 56.41 | 49.11 | 51.11 | 48.67 | 45.83 | 55.75 | 57.46 | 28.56 | 24.07 | 25.39 | 23.68 | 21.79 | 30.15 |
| | TRADES [45] | 81.93 | 57.49 | 52.66 | 53.68 | 50.58 | 49.23 | 57.59 | 55.23 | 30.48 | 27.79 | 28.53 | 25.06 | 23.94 | 31.83 |
| | ARD [14] | 83.93 | 59.31 | 52.05 | 54.50 | 51.22 | 49.19 | 58.36 | 60.64 | 33.41 | 29.16 | 30.30 | 27.85 | 25.65 | 34.50 |
| | IAD [52] | 83.24 | 58.60 | 52.21 | 54.18 | 51.25 | 49.10 | 58.09 | 57.66 | 33.26 | 29.59 | 30.58 | 29.37 | 25.12 | 34.26 |
| | RSLAD [53] | 83.38 | 60.01 | 54.24 | 55.94 | 53.30 | 51.49 | 59.72 | 57.74 | 34.20 | 31.08 | 31.90 | 28.34 | 26.70 | 34.99 |
| Data Free | DAFL[5] | 54.98 | 27.04 | 24.75 | 25.87 | 22.90 | 22.25 | 29.63 | 41.67 | 21.42 | 20.13 | 20.81 | 17.96 | 17.16 | 23.19 |
| | DFAD [12] | 57.58 | 31.54 | 29.68 | 30.65 | 26.94 | 26.47 | 33.81 | 37.57 | 18.95 | 17.53 | 18.14 | 15.06 | 14.57 | 20.30 |
| | ZSKT [28] | 58.08 | 31.98 | 29.94 | 30.92 | 27.21 | 26.68 | 34.13 | 38.91 | 20.16 | 18.78 | 19.41 | 16.38 | 15.52 | 21.52 |
| | CMI [13] | 53.28 | 25.78 | 23.14 | 23.97 | 21.03 | 20.38 | 27.92 | 45.04 | 22.78 | 21.02 | 21.90 | 17.90 | 16.97 | 24.26 |
| | DFARD [37] | 66.44 | 38.53 | 35.94 | 37.15 | 32.79 | 32.14 | 40.49 | **46.33** | 24.56 | 22.94 | 23.59 | 20.12 | 19.19 | 26.12 |
| MSA | DFHL_RS [43] | **77.86** | 44.94 | 40.07 | − | 40.64 | **39.51** | 48.60 | 51.94 | 23.68 | 20.02 | 19.88 | 20.91 | 19.30 | 25.95 |
| | **DERD (Ours)** | 72.83 | **62.32** | **53.64** | **54.01** | **53.71** | 36.03 | **55.42** | 40.21 | **27.26** | **25.94** | **26.07** | **25.88** | **21.39** | **27.79** |

**Table 3: White-box robustness evaluation on MobileNet2 for CIFAR10 and CIFAR100. The best results obtained through data-free methods are highlighted in bold.**

| | Models | CIFAR10 | | | | | | | CIFAR100 | | | | | | |
|---|---|---|---|---|---|---|---|---|---|---|---|---|---|---|---|
| | | Clean | FGSM | $PGD_S$ | $PGD_T$ | CW | AA | Ave | Clean | FGSM | $PGD_S$ | $PGD_T$ | CW | AA | Ave |
| | Nature | 92.95 | 14.47 | 0.0 | 0.0 | 0.0 | 0.0 | 17.90 | 74.58 | 7.19 | 0.0 | 0.0 | 0.0 | 0.0 | 13.62 |
| Data Driven | SAT [27] | 83.38 | 56.41 | 49.11 | 51.11 | 48.67 | 45.83 | 55.75 | 56.85 | 31.95 | 28.33 | 29.5 | 26.85 | 24.71 | 33.03 |
| | TRADES [45] | 81.93 | 57.49 | 52.66 | 53.68 | 50.45 | 49.23 | 57.57 | 56.20 | 31.37 | 29.21 | 29.83 | 25.06 | 24.16 | 32.63 |
| | ARD [14] | 83.93 | 59.31 | 52.05 | 54.20 | 51.22 | 49.19 | 58.31 | 59.83 | 33.05 | 29.13 | 30.26 | 27.86 | 25.53 | 34.27 |
| | IAD [52] | 83.24 | 58.60 | 52.21 | 54.18 | 51.25 | 49.10 | 58.09 | 56.14 | 32.81 | 29.81 | 30.73 | 27.99 | 25.74 | 33.87 |
| | RSLAD [53] | 83.38 | 60.01 | 54.24 | 55.94 | 53.30 | 51.49 | 59.72 | 58.97 | 34.03 | 30.40 | 31.36 | 28.22 | 26.12 | 34.85 |
| Data Free | DAFL[5] | 47.53 | 24.51 | 21.18 | 22.09 | 19.50 | 18.86 | 25.61 | 40.46 | 20.63 | 19.03 | 19.78 | 16.54 | 15.82 | 22.04 |
| | DFAD [12] | 56.13 | 29.73 | 26.48 | 27.64 | 24.35 | 24.02 | 31.39 | 25.41 | 12.75 | 11.42 | 11.95 | 9.58 | 9.24 | 13.39 |
| | ZSKT [28] | 57.02 | 30.29 | 27.07 | 28.25 | 24.89 | 24.40 | 31.98 | 26.16 | 12.34 | 11.36 | 11.78 | 9.69 | 9.16 | 13.41 |
| | CMI [13] | 44.53 | 21.34 | 19.67 | 19.97 | 16.25 | 15.97 | 22.95 | 40.23 | 19.76 | 17.96 | 18.56 | 14.86 | 14.02 | 20.89 |
| | DFARD [37] | 61.16 | 34.46 | 31.66 | 32.80 | 28.40 | 27.90 | 36.06 | **41.78** | 22.04 | 20.84 | 21.68 | 17.93 | 17.04 | 23.55 |
| | **DERD (Ours)** | **64.28** | **60.91** | **47.52** | **48.04** | **50.31** | **34.88** | **50.99** | 32.12 | **24.61** | **24.89** | **24.96** | **25.01** | **18.13** | **24.95** |

epochs, with the learning rate reduced by a factor of 0.1 at epochs 800 and 1600. Madry's AT [27] is used to train the robust teacher. For the student models, stage-I includes 2000 epochs of natural training, followed by stage-II, which consists of 100 epochs of robustness training. The initial learning rates for CIFAR10 and CIFAR100 are set to 0.01. For ImageNet100, we fine-tune pre-trained models on ImageNet as both the nature teacher and robustness teacher, with an initial fine-tuning learning rate of 0.0001. Table 1 present the evaluation results for supplementary materials the teacher models. Please refer to the suppMore details of the experiments

## 4.2 Experimental Results

**White-box robustness.** Table 2 and Table 3 present the experimental results of the white-box attacks on CIFAR10 and CIFAR100. We report the results of both data-driven methods and the direct adaptation of several existing data-free distillation methods to robustness

distillation. We also reported the results of DFHL_RS [43] in Table 3, considering that model robust stealing attack (MSA) can serve as a special data-free robustness distillation. Note that DFHL_RS is only evaluated on ResNet in its original experiments. The results of existing methods are obtained from previous literature [37, 43, 53]. The results on ImageNet100 are moved to the supplementary for sapce reasons. Intuitively, our DERD demonstrates significant superiority compared to directly applying existing data-free distillation methods to robustness distillation, and it is comparable to data-driven robustness distillation methods. However, despite adopting a teacher group-based expert strategy to optimize the trade-off, the accuracy of the student on clean samples remains significantly lower than that of data-driven methods. The conflict between robustness and accuracy is undoubtedly amplified in the absence of real data. Nonetheless, our DERD brings reliable robustness to the

**Table 4: Black-box robustness on CIFAR10.**

| Methods | ResNet-18 | | | MobileNetV2 | | |
|---|---|---|---|---|---|---|
| | $PGD_S$ | CW | Square | $PGD_S$ | CW | Square |
| SAT | 60.84 | 60.52 | 54.27 | 60.46 | 59.83 | 53.94 |
| TRADES | 62.20 | 61.75 | 55.13 | 60.90 | 60.23 | 53.46 |
| RSLAD | 64.11 | 63.84 | **57.90** | 63.30 | 63.20 | 56.70 |
| **DERD (ours)** | **67.83** | **66.76** | 56.03 | **64.16** | **64.84** | **57.01** |

**Table 5: Ablation analysis of our DERD on CIFAR10.**

| | Modules | Clean | FGSM | PGDS | PGDT | CW | AA | Ave. |
|---|---|---|---|---|---|---|---|---|
| | +stage-I | 91.87 | 22.24 | 11.54 | 12.01 | 10.41 | 6.79 | 25.79 |
| +stage-II | +SA | 70.14 | 52.13 | 38.24 | 39.68 | 37.99 | 28.62 | 44.64 |
| | +SA & EG | 72.03 | 65.04 | 52.89 | 53.62 | 53.81 | 35.16 | 55.42 |
| | +EG & SA & SGA | 72.83 | 67.39 | 53.64 | 54.04 | 53.01 | 36.03 | 56.29 |

student model without real data, and the trade-off can be mitigated by the regularizer, as demonstrated in the ablation analysis later.

**Black-box robstness.** Following the RSLAD [53] setting, we also conduct a brief evaluation of the black-box robustness of our DERD on CIFAR10. We use ResNet50 to create adversarial samples of PGD and CW attacks for transfer-based attacks, and square attack for query-based attacks. The attack budgets are consistent with those used for white-box attacks. The experimental results are presented in Table 4. Since there is a lack of black-box evaluation for data-free robustness distillation, we compare DERD with several common data-driven adversarial training and robustness distillation methods. Notably, our DERD achieves comparable black-box robustness, demonstrating the transferable robustness.

### 4.3 Ablation Study

**Ablation of the modules.** We conduct an ablation study to evaluate the incremental effects each module in DERD on CIFAR10 with ResNet-18 as the backbone. The results are summarized in Table 5. The complete DERD includes stage-I and stage-II, while stage-II includes modulus of EG, SA and SGA. Note that when SA strategy works alone, DERD degenerates into training the student models by using the discrepancy data $x_d$ and the sole robust teachers, like Eq. 12, where $T_{nat}$ is replaced by $T_{rob}$. Intuitively, stage-I can be considered as the pre-training process to obtain the natural knowledge. The SA module effectively improves the robustness of the student model. The EG and SGA strategy comprehensively enhance the model's robustness and accuracy by promoting gradient harmony and augmentation. The results of the ablation analysis align with our expectations for the modules and highlight the importance of both SA strategy and EG / SGA module in improving the overall performance of our DERD.

**Without stage-I?** Directly distilling robust knowledge without stage-I yields suboptimal results in terms of both accuracy and adversarial robustness. To verify this, we conduct verification experiments on CIFAR10, and the results are shown in Table 6. The results indicate that stage-I brings a significant increment to DERD. The suboptimal robustness without stage-I may stem from two main reasons. First, generating pseudo adversarial data relies on pseudo nature data (refer to Eq. 13). The process of generating pseudo data

**Table 6: Ablation analysis of stage-I on CIFAR10. Without stage-I, DERD directly optimizes the randomly initialized generator and student.**

| | Clean | FGSM | $PGD_S$ | $PGD_T$ | CW | AA | Ave |
|---|---|---|---|---|---|---|---|
| w/o stage-I | 30.12 | 41.67 | 33.26 | 33.11 | 32.87 | 24.57 | 32.60 |
| **DERD (ours)** | **72.83** | **67.39** | **53.64** | **54.01** | **53.71** | **36.03** | **56.29** |

for adversarial samples requires that the generator $G$ is already capable of producing pseudo natural data. Subsequently, both the generator and the student model can be further optimized based on this foundation. Secondly, it could be challenging for the student to directly acquire robust knowledge. However, initializing the student with natural knowledge can facilitate more efficient learning of robust knowledge. According to ARREST [34], pre-training on natural knowledge can lead to more stable representations during robustness training. Therefore, it is necessary to introduce stage-I as a warm-up for both the generator $G$ and the student model $S$.

## 5 DISCUSSION

We briefly discuss the expansibility of DERD from two perspectives.

**Extension to model inversion framework.** The model-inversion-based method is also a main paradigm of data-free distillation. Our DERD can be extended to this framework as an alternative solution. In this scenario, the explicit generator $G$ becomes implicit, where the input noise tensor is directly optimized.

**Handling absence of natural teacher.** While DERD relies on the presence of both a natural teacher and a robust teacher, we propose an alternative approach for scenarios where only a robust teacher is available. We find that student tends to first learn the natural knowledge before acquiring robust knowledge, making itself a good surrogate for the natural teacher.

For detailed discussions on these issues, please refer to the supplementary materials. In summary, while the alternative solutions can achieve certain accuracy and adversarial robustness, the complete DERD demonstrates significant superiority. This is attributed to the controllability of the explicit generator model and the discriminative nature knowledge provided by nature teachers.

## 6 CONCLUSION AND OUTLOOK

We consider the challenge of distilling the robustness from high-performance large models to high-efficiency small models without the real data, and propose Data-free Experts-guided Robustness Distillation (DERD), where a novel dual-level adversarial learning mechanism and an efficient stochastic gradient aggregation module are proposed. Experimental results corroborate that DERD is superior to existing attempts of data-free robustness distillation, and can even achieve robustness comparable to data-driven robustness distillation. Still, DERD relies on a strong assumption of the dual-teacher hypothesis. Although effective, the concurrent requirement for both a robust teacher and a natural teacher may introduce additional memory costs and privacy threats. Furthermore, models distilled in a data-free paradigm sometimes suffer from unstable convergence, which is also a potential improvement direction.

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
