# OpenReview forum: "DERD: Data-free Adversarial Robustness Distillation through Self-adversarial Teacher Group"
_acmmm.org/ACMMM/2024/Conference — MM2024 Poster_

### Official Review · Reviewer_57zN · 2024-05-24

**Rating:** 4
**Confidence:** 3

**Summary:**

This paper proposed a dual-level data-free robustness distillation method that can distill robustness to a student model without real data. There exist two stages in the strategy including a warm-up stage for the generator and student model, and a stage to distill robustness and accuracy. Experiments on CIFAR10, CIFAR100 and ImageNet100 are conducted to demonstrate the performance of the method.

**Strengths:**

1. The paper is well written and easy to follow.
2. The experimental results demonstrate the performance.

**Limitations:**

Several questions need to be addressed:

1. How are the feature statistics of real data acquired in Eq. 8 if the condition is data-free? Did you use the statistical data stored in the batch normalization layers of the teacher model? Were all BN layers involved in Eq. 8?

2. The paper maximizes the output discrepancy between $T_{nat}$ and $T_{rob}$ to create pseudo adversarial data rather than maximizes the loss of $T_{nat}$ or $T_{rob}$ or the student model like AT does. Is there any reason for it and have you tried to train the student model by AT with pseudo data?

3. The paper provides the results on CIFAR10 and CIFAR100. The results on ImageNet100 are in supplementary material. It would be better if the authors can run some comparison methods on ImageNet100 to demonstrate the performance.

4. I wonder if AT is time consuming on ViT. The paper mentioned the robust model is trained for 2000 epochs, what is the training time?

Minor:
1. Probably the equation referenced in Line 630 should be Eq.11 rather than Eq. 10?

**Suitability:**

3

---

### Official Review · Reviewer_Fcm2 · 2024-05-25

**Rating:** 4
**Confidence:** 2

**Summary:**

This paper proposes a robustness distillation method called DRED for scenarios where original data is inaccessible or absent. In general, DRED consists of two stages: a warm-up stage to train a generator for the generation of pseudo data, and a distillation stage for distilling robustness&accuracy from corresponding teacher models to student models.

**Strengths:**

1. Extensive experimental results validate the effectiveness of DRED in data-free scenarios, surpassing the existing method w.r.t. robustness by a considerable amount.
2. Despite the complexity of DRED, the authors present their methods clearly and logically, making them easy to follow and understand. Illustrations are also intuitively understandable.
3. Experiment designs are reasonable, validating the effectiveness of major modules employed in the paper. Discussion and analysis are also sufficiently conducted.

**Limitations:**

I believe the general quality of the paper is good, but I do have a few concerns/questions:

1. The paper, despite its soundness and solidity, seems to care only about visual tasks, without discussion of any other modality, which might not align with the interest of MM.

2. Stage I seems to be completely driven by the natural teacher model, without justification for the genuine quality of the generated pseudo data. This could be less robust as it relies very much on how well the teacher model captures the distribution of real data, and may easily lead to overfitting. Supplementary docs have confirmed the visual inconsistency between the generated data and real data. Would adopting external supervision on the quality of generated data further boost the performance of DRED?

3. In the self-adversary stage, the author uses discrepancy data xd for generating adversarial samples. Could the authors please elaborate on how exactly those discrepancy data xd were constructed? I did not find any further details about it.

4. Besides, as mentioned in the paper, the generation of adversarial samples is unbounded and guided by maximizing the discrepancies. Does the optimization incorporate label information to maximize the mislabeling probability? If No, would incorporating label information to guide self-adversary benefit DRED w.r.t. robustness?

5. Training a generator could be time-consuming and challenging, considering the difficulty in convergence for GAN. This could be more problematic for larger and more diverse datasets such as ImageNet. According to the appendix, the result of DRED on ImageNet is not as promising as CIFAR, significantly surpassed by SAT. This could also be the performance degradation because of the quality of generated data.

**Suitability:**

2

---

### Official Review · Reviewer_exck · 2024-05-27

**Rating:** 3
**Confidence:** 4

**Summary:**

This study investigates the concept of data-free distillation. The authors argue that most current robustness distillation approaches rely heavily on data, making them unsuitable for deployment in data-sensitive environments. Furthermore, they highlight that the trade-off between robustness and accuracy often persists from the teacher model to the student model, and there has been a lack of discussion on addressing this trade-off within a data-free context. In response to these challenges, the authors introduce a novel Data-free Experts-guided Robustness Distillation (DERD) technique to extend robustness distillation to scenarios where data is not available. Experimental results demonstrate the effectiveness of the proposed method.

**Strengths:**

1. This paper explores data-free adversarial distillation, which has important implications for secure deployment of small models based on privacy considerations.
2. The proposed method implements min-max game on DFARD.
3. This paper attempts to explore the trade-off of accuracy and robustness on DFARD.

**Limitations:**

1. As a rigorous researcher, we generally use "adversarial example" instead of "adversarial sample".
2. Some descriptions of related work are inaccurate. DeepFool [29] is an optimized white-box attack.
3. The motivation for the approach is not very clear. This paper introduces a lot of loss function terms, and their motivations are very vague. We believe that the relationship with the challenges of this paper needs to be re-described. In addition, many loss functions are based on existing work, which limits novelty.
4. Ablation experiments for training goals are lacking. The total loss function has 8 sub-terms, which is very difficult to train and determine the parameters. The corresponding coefficients should appear in the main body for reproducibility. Table 5 only provides stage-level ablation, and we do not know whether each sub-item is really effective, especially in Equation 11. In addition, how are the corresponding detailed parameters determined?
5. In addition, how is the trade-off of accuracy and robustness reflected? There is no relevant experiment to explain in the main body.

**Suitability:**

2

---

### Official Review · Reviewer_4aZg · 2024-05-27

**Rating:** 4
**Confidence:** 2

**Summary:**

This paper proposes Data-free Experts-guided Robustness Distillation, which facilitates robustness distillation under the data-free setting. This is achieved by a dual-level adversarial distillation under the guidance of two teachers, a robust teacher and a nature teacher. The paper deploys a two-stage training to learn a robust student. Stage 1 warms up the generator through an adversarial training between the student and the generator. Then in the second stage, a fixed robust teacher is introduced to learn the robustness in students, while preserving a good Acc.

**Strengths:**

`. The two-stage design proposed in this paper is particularly intriguing. The experimental results presented in Table 3 demonstrate that the method effectively learns a robust model, outperforming other existing methods. This highlights the potential of the two-stage approach in enhancing model performance and robustness.

2. The authors have conducted comprehensive experiments on both CIFAR-10 and CIFAR-100 datasets, employing a variety of attack methods to evaluate the resilience of the proposed model. These experiments provide valuable insights into the effectiveness of the method under different adversarial scenarios.

**Limitations:**

1. The method has been evaluated solely on CIFAR, which is a relatively small dataset for comprehensive evaluation. To enhance the credibility of the results, it would be beneficial for the authors to include evaluation outcomes on ImageNet-256 or a subset of ImageNet-256.
2. The paper demonstrates that stage 1 is crucial for developing robust student models. However, it lacks an in-depth analysis of the underlying reasons. Additional analytical results are needed to substantiate this assertion.

**Suitability:**

3

---

### Official Review · Reviewer_Mgkz · 2024-06-01

**Rating:** 3
**Confidence:** 4

**Summary:**

This paper proposed a two stage data-free distillation method to improve the adversarial robustness. While previous work DFARD applies distillation to a teacher model with generated adversarial samples, this paper introduces an additional robustness teacher model for better performance.

**Strengths:**

Limit research has been done on the field of improving adversarial robustness with no access to the original benign data. This paper builds their work based on DFARD. In DFARD, distillation is applied to a teacher model with generated adversarial samples. This submission proposed adding a robust teacher for better adversarial samples generation and better distillation process. To make the distillation of clean and robust teacher work, a warmup stage is also proposed. Compared with other data-free methods, in majority of the cases, their method achieves higher performance on both clean and adversarial samples.

**Limitations:**

In the third contribution, the authors claim that their performance is comparable to the data-driven distillation methods which is not true according to table 2 and table 3. In table 2 and 3, we can clearly see that their proposed methods have much lower clean accuracy than all data driven methods, and have lower robustness under autoattack than all data driven methods. This contribution is overclaimed.

Even plenty of results are shown in the submission, lots of details are missing and
From Table 6, we can see the importance of stage 1. However, the loss components in stage 1 are mostly designed on certain assumptions and with no support of solid ablation study or results.  Details about how the other methods’ performance are collected are missing. This might bring up concern about the fairness of benchmarking with other methods. These details can’t be found in the main context, the supplement nor the attached codes. While generative samples are one of the key parts, no generated samples are revealed, no quality analysis are done. How the robust teacher is trained is also unknown. We only know that adversarial training is applied, but have no idea of the attack budget, iteration. It would be interesting if the authors can show how the generated adversarial samples would perform on clean and robust teachers. How different robust teachers would affect the performance is also worth exploring.

**Suitability:**

3

---

### Meta-Review · Area_Chair_V5Wz · 2024-07-02

**Recommendation:** Accept (Poster)
**Confidence:** 4

**Metareview:**

This paper presents a novel approach for robustness distillation in scenarios where original data is inaccessible. The proposed method involves a warm-up stage to train a generator for creating pseudo data, followed by a distillation stage to transfer robustness and accuracy from teacher to student models. The experiments demonstrate the effectiveness of DERD on CIFAR10, CIFAR100, and ImageNet100 datasets.

Pros:
- This approach does not require access to original training data which expands the applicable areas for adversarial training methods.

- Empirical evaluations are extensive. Considerable improvements are observed compared with existing techniques in data-free scenarios.

Cons:

- Reviewers have concerns over the hyper-parameters and other training details. The missing details make this paper less reproducible. Thus, it is recommended to release source code.

- Some reviewers question about the technical novelty of this work which looks like a combination of multiple existing losses.

Overall, this  work tackles a challenging task of adversarial training without access to source training data. The contributions to the field of robustness distillation in data-free scenarios are sufficient to be accepted by ACM MM.